# Microstructure and Corrosion Behavior of Laser-Welded Al–Mn–Zr Alloy for Heat Exchanger

**DOI:** 10.3390/ma16176009

**Published:** 2023-09-01

**Authors:** Jeong-Min Lim, Yoon-Sik So, Jung-Gu Kim

**Affiliations:** School of Advanced Materials Science and Engineering, Sungkyunkwan University, 2066, Seobu-ro, Jangan-Gu, Suwon-si 16419, Gyeonggi-do, Republic of Koreasoy4718@skku.edu (Y.-S.S.)

**Keywords:** laser welding, aluminum alloy, corrosion, zirconium, fusion zone, micro–galvanic, interdendritic segregation, passive film

## Abstract

In this study, an Al–Mn–Zr alloy was designed and its microstructure and corrosion behavior compared after laser welding to that of AA3003. As the results of immersion and electrochemical tests showed, both alloys had a faster corrosion rate in the fusion zone than in the base metal. Laser welding caused interdendritic segregation, and spread the intermetallic compounds (IMCs) evenly throughout in the fusion zone. This increased the micro-galvanic corrosion sites and destabilized the passive film, thus increasing the corrosion rate of the fusion zone. However, Zr in the Al–Mn alloy reduced the size and number of IMCs, and minimized the micro-galvanic corrosion effect. Consequently, Al–Mn–Zr alloy has higher corrosion resistance than AA3003 even after laser welding.

## 1. Introduction

A heat exchanger is a core part that directly affects the efficiency of the air conditioner and ventilation system. Fin–tube heat exchangers are generally used [1]. As a material for the fin and tube, AA3003 (Al–Mn alloy) is used due to the advantage of light weight and good formability as well as high corrosion resistance [2,3]. Brazing was used to join the Al fin and tube, but more recently welding has been used. The brazing melts the filler metal and joins the base metals; hence, the use of filler metal is essential. However, the filler metal can cause heat exchanger failure by causing galvanic corrosion between the filler metal, fin, and tube [4,5,6]. Welding, on the other hand, does not necessarily use filler metal, and can be used as an alternative joining method. In particular, laser welding is attracting attention as a fast and accurate fully automatic process with a narrow heat–affected zone (HAZ) and weld seam [7,8]. However, it has been reported that the microstructural changes caused by laser welding make the fusion zone susceptible to pitting corrosion [9,10,11,12,13,14,15,16,17]. Therefore, to improve a lifespan of fins and tubes, it is necessary to study the effects of laser welding on the microstructure and corrosion behavior of Al–Mn alloys.

In Al–Mn alloys, the Mn element is precipitated as Al_6_Mn IMCs. However, in most Al alloys, the Fe element is contained as an impurity and exists as Al_3_Fe and Al_6_Fe intermetallic compounds (IMCs). Al_6_Fe IMC combines with Al_6_Mn to form Al_6_(Fe,Mn), which has similar corrosion potential to the Al matrix and reduces the micro-galvanic corrosion between IMCs and the Al matrix [18,19]. However, the remaining Al_3_Fe and Al_6_Fe have higher corrosion potentials than the Al matrix and still act as cathodic sites, accelerating the micro-galvanic corrosion of Al–Mn alloys. Therefore, in order to increase the corrosion resistance of Al–Mn alloys, studies on the uniform distribution of IMCs by adding alloying elements or nanoparticles are being conducted [20]. Zr is known as an alloying element for refining the grain and IMCs. Zr in Al alloys precipitates as fine Al_3_Zr, which prevents the grain growth [21,22,23]. In addition, the melting temperature of Al_3_Zr is very high, thus some Al_3_Zr remains unmelted in the molten pool during the welding [24]. These fine Al_3_Zr particles can play the same role as nanoparticles. The addition of nanoparticles reduces the IMC size and spreads the IMC of the welded and base Al alloy [25]. Therefore, the result of increasing the corrosion resistance of the fusion zone can be expected through the adding of Zr.

In this study, an Al–Mn alloy containing the Zr was prepared to estimate the effect of Zr on the corrosion behavior of laser-welded Al–Mn alloy, and the corrosion behavior of Al–Mn–Zr alloy was compared with that of AA3003. The microstructural change during the laser welding was observed using optical microscopy (OM), scanning electron microscopy (SEM), and electron probe microanalysis (EPMA). Then, the corrosion behavior was estimated by immersion test, potentiodynamic (PD) test, and electrochemical impedance spectroscopy (EIS).

## 2. Materials and Methods

### 2.1. Materials and Laser Welding Process

The Al alloys used in this research were commercial AA3003 and Al–Mn–Zr alloys (UniCorAl^®^, Suwon, Republic of Korea, U3003). U3003 is an Al–Mn alloy with added Zr which was developed to improve corrosion resistance. The U3003 alloy was prepared using a 99.9% Al ingot and three master alloys (Al–20Mn, Al–5Fe, and Al–15Zr). An appropriate amount of material was filled into a graphite crucible at 780–800 °C, then the molten metal was injected into a mold and air cooled. Following the casting, hot and cold rolling were performed to produce a 1 mm-thick plate. The chemical compositions of the two alloys were analyzed by inductively coupled plasma–mass spectrometry (ICP–MS, Perkin Elmer (Waltham, MA, USA), NexION 2000S), as shown in Table 1.

The laser-welded specimens were plate shaped with a size of 60 mm × 20 mm × 1 mm, and the center was welded in the longitudinal direction. Figure 1 shows the specimen preparation procedure.

Dual-beam laser welding was performed in a butt joint configuration with no filler metal. Pure nitrogen gas was used as the shielding gas with a flow rate of 10 L/min. The welding parameters are listed in Table 2.

### 2.2. Surface Analysis

To observe the microstructure of the specimens, the specimens were cut perpendicularly to the welding direction and mounted using epoxy resin. The cross sections of the specimens were mechanically polished up to 4000 grit using SiC paper, followed by polishing to 1 μm using alumina powder. Before observation, the specimens were etched using a Keller reagent (5 mL HNO_3_, 3 mL HCl, 2 mL HF, and 190 mL deionized water) according to ASTM E407 [26]. The microstructure of the laser welded specimens was characterized by OM (Witec (Ulm, Germany), Alpha 300S) and SEM (JEOL (Akishima-shi, Japan), JSM 6380). Their chemical compositions were measured by EPMA (JEOL, JXA 8900R) equipped with SEM.

### 2.3. Corrosion Testing

To evaluate the corrosion behavior of laser-welded Al–Mn alloys, immersion, potentiodynamic (PD), and EIS testing were performed. The test solution was artificial acid rain, and the composition is presented in Table 3 [27]. The ion concentration of the test solution was controlled by H_2_SO_4_, HNO_3_, and NaCl, and the pH was adjusted with 0.1 M NaOH solution.

For the immersion test specimens, the top weld side was polished to a 1000-grit size using SiC paper. Then, the specimen was immersed in the solution for 12 weeks. During the immersion test, the solution was changed every 2 weeks to maintain the chemical composition. After the immersion test, the corrosion depth was measured using a surface profiler (KLA Tencor (Milpitas, CA, USA), Alpha Step 500).

The electrochemical tests were conducted on the bare Al–Mn alloy (before laser welding) and the fusion zone of laser-welded specimens, respectively, using a three-electrode system and a potentiostat (Bio Logic (Seyssinet-Pariset, France), VSP 300). The working electrode was the test specimen, the counter electrode was the pure carbon rod, and the reference electrode was the saturated calomel electrode. Each electrochemical test was performed in triplicate, and before conducting electrochemical tests, the open circuit potential (OCP) was measured for 30 h. The PD tests were performed with a scan rate of 0.166 mV/s, following the ASTM G5 standard [28]. The anodic potential was swept from the OCP to 1 V_OCP_, while the cathodic potential was swept from OCP to −0.6 V_OCP_. The frequency range of the EIS tests was from 100 kHz to 10 MHz with alternating current amplitude of 20 mV.

## 3. Results and Discussion

### 3.1. Microstructure of Laser-Weld Beads

Figure 2 shows the cross-sectional images of laser-welded AA3003 and U3003.

The red boxes in Figure 2a,c indicate the interface between the base metal (BM) and the fusion zone (FZ), and enlarged images are shown in Figure 2b,d. In the FZ, the grains grow epitaxially from the BM/FZ boundary towards the weld center, which is in a perpendicular direction to the welding direction. Unlike in the BM, the distribution of IMCs in the FZ exhibits directionality, clearly distinguishing BM and FZ. Since the laser welding enables rapid melting and cooling, the effect of heat on the BM is reduced [29,30]. Therefore, both alloys have a very narrow HAZ. Consequently, in this study, the cross sections of the laser-welded specimens were divided into two regions, BM and FZ.

Figure 3 shows the SEM images of the BM/FZ interface (red boxes in Figure 2), and the distribution of Fe and Mn at the same location.

In both alloys, Fe and Mn were concentrated in the IMC, indicated by the white circle in the BM, whereas they were dispersed over the entire area in the FZ. Consequently, the laser welding affected the microstructure of FZ, especially the IMC distribution. Figure 3c,d show the EDS point analysis results of the IMC inside the red circle for each alloy. The IMCs of both alloys contained similar amounts of Fe and Mn, with a significant presence of Al. Therefore, the IMC shown in the SEM image is Al_6_(Fe,Mn), which is a representative IMC of Al–Mn alloy [31,32]. Since Al_6_(Fe,Mn) is formed by substituting a Fe atom at the Mn position of Al_6_Mn or by substituting a Mn atom at the Fe position of Al_6_Fe, Al_6_(Fe,Mn) has a full composition range from Al_6_Mn to Al_6_Fe [33,34]. Al_6_Mn has a similar potential with the Al matrix, then increasing the corrosion resistance. However, Al_6_Fe has a higher corrosion potential than Al matrix, of about 300 mV, thus Al_6_Fe acts as a cathode to the Al matrix and accelerates the corrosion rate of the Al matrix by the micro-galvanic corrosion [19,35]. Therefore, the distribution of IMC affects the corrosion behavior of Al–Mn alloy. Since the micro-galvanic corrosion rate increases as the number of micro-galvanic sites and the area ratio of the cathode to the anode increase, the IMC size and distribution should be analyzed.

The microstructures of each region are shown in Figure 4.

In Figure 4a–c, the IMC in the BM (solid arrows) of AA3003 is a short-rod shape with an average size of 1.282 μm^2^ and the IMCs are distributed without regularity. The IMCs in the FZ of AA3003 appear as very thin bands along the grain growth direction (dotted arrows). In Figure 4d–f, the IMCs in the BM of U3003 are spherical with an average size of 0.291 μm^2^ and are distributed irregularly. However, in the FZ, the size of IMCs is smaller than in BM and distributed along the grain growth direction. This means that during the laser welding, the alloying elements in the molten pool were accumulated among the dendrite arm and interdendritic segregation occurred in both alloys [36,37]. Moreover, in Figure 4c, some equiaxed dendrites appear at the weld center of AA3003, further showing the interdendritic segregation.

Additionally, if the cooling rate of the laser-welding process is fast, then the dendrite arm spacing is reduced [38]. Consequently, the interdendritic IMCs in the dendrite arms are smaller and more evenly spread.

During the solidification, Zr in Al alloy is precipitated as fine Al_3_Zr particles, and this precipitation process hinders the nucleation and growth of IMCs [20,39]. Therefore, in BM, the IMC size of U3003 is smaller than that of AA3003. Additionally, after laser welding, since the melting point of Al_3_Zr is significantly higher (1500 °C) than that of Al (660 °C), some Al_3_Zr particles remain unmelted [24,40]. These fine Al_3_Zr particles act as a heterogeneous nucleation site of Al_6_(Fe,Mn) IMCs. Consequently, a large number of fine spherical IMCs are formed.

Figure 5 shows the area ratio of IMCs to the Al matrix calculated from the SEM images of Figure 4.

In the AA3003, the area ratio of IMCs to Al matrix is 6.21% in BM, and 6.81% in FZ. In the U3003, the area ratio of IMCs is 1.54% in BM and 1.58% in FZ. Since the rapid cooling of laser welding refines and spreads the IMC, the area ratio of the IMC in the FZ of both alloys increases [41]. In FZ of AA3003, the IMCs are band shaped and are spaced close to each other by the interdendritic segregation; thus, the area of IMC becomes 9.7% wider than that of BM. However, in the case of U3003, Zr in the Al alloy hindered the growth of Al_6_(Fe,Mn) and made the IMC spherical, resulting in an increase of the IMC area by only 2.6%. Since U3003 has a lower content of Fe and Mn elements than AA3003, the amount of IMC is less in U3003 in both regions. The galvanic corrosion rate is proportional to the area ratio of the cathode to the anode; thus, the corrosion rate of the Al–Mn alloy is increased as the IMC area increases [42,43]. Therefore, it can be inferred that the corrosion rate of U3003 is lower than that of AA3003, in both regions.

### 3.2. Corrosion Testing

To investigate the corrosion behavior of laser-welded specimens, immersion testing was performed for 12 weeks. The surface and cross-sectional images of specimens after the immersion test are shown in Figure 6.

In case of AA3003 (Figure 6a,b), the whole region of AA3003 was corroded, and the maximum corrosion depth of FZ was 61.3 µm, which was more severe than that of BM. In Figure 6c,d, it can be seen that U3003 was corroded very slightly in BM, and the partial area of FZ was corroded with a maximum corrosion depth of 18.5 µm. Therefore, the corrosion rate of AA3003 was faster than that of U3003 in both regions.

Figure 7 shows the polarization curves of each specimen and the results are presented in Table 4.

Consistent with the results of the immersion test, the corrosion rate of BM was faster than that of the FZ in both alloys. However, the corrosion potential difference of BM and FZ was less than 5 mV. Therefore, the galvanic corrosion by the microstructure difference between the BM and FZ was not significant. Consequently, the microstructure change caused by the laser welding affected the micro-galvanic corrosion between the IMC and matrix.

Figure 8 shows the high-magnification images of the cross sections after immersion testing. In the BM of AA3003 (Figure 8a), the corrosion of the Al matrix was concentrated around the IMC.

This means that the micro-galvanic corrosion occurred between the Al_6_(Fe,Mn) IMC and Al matrix. In contrast, in FZ (Figure 8b), corrosion occurred uniformly over the entire area. Since the IMCs were refined and redistributed in FZ after laser welding, the number of micro-galvanic sites increased. Therefore, corrosion occurred uniformly over a wide area without being concentrated, and the micro-galvanic rate of FZ was increased compared to that of BM. In the case of U3003 (Figure 8c,d), the IMC size was quite small, and micro-galvanic corrosion next to the IMC did not appear. Since Al_6_(Fe,Mn) has a composition between Al_6_Mn and Al_6_Fe, the corrosion potential also has a value between that of Al_6_Mn and Al_6_Fe. As the Mn content increased in IMC, the corrosion potential decreased from that of Al_6_Fe to that of Al_6_Mn, which had a corrosion potential similar to that of the Al matrix [19,35]. The Mn/Fe ratio of AA3003 is 3.15 and that of U3003 is 5.6. Since the Mn content of IMC increases with the Mn/Fe ratio, the corrosion potential of Al_6_(Fe,Mn) IMC is lower in U3003 than in AA3003. Therefore, the micro-galvanic effect was not significant in U3003.

In the previous section, the area ratio of IMC to the Al matrix of BM and FZ only slightly increased in U3003. In addition, no significant micro-galvanic corrosion was observed. Nevertheless, the corrosion depths of BM and FZ were quite different, as shown in Figure 6. There is another reason for the corrosion acceleration of FZ in addition to micro-galvanic corrosion between the IMC and Al matrix. In Figure 8c, the result of the EDS line profile performed at the red dashed arrow is shown in the lower left corner. The scan direction was inward from the alloy surface. The oxygen content decreases and Al content increases along the scan direction. Therefore, it can be found that the oxide film was formed on the specimen surface [44,45]. In Figure 8c, the oxide film becomes thin on the corroded surface. Moreover, in Figure 8d, the IMCs are appearing in the passive film. The breakdown of passive film is the main mechanism of pitting corrosion of Al [46]. Therefore, in this study, the stability of oxide film in each region was investigated through the EIS test.

To investigate the corrosion mechanism of laser-welded Al–Mn alloys, the EIS tests were performed on FZ and BM, and the impedance plots were analyzed based on an appropriate equivalent circuit using ZSimpWin Software version 3.21. Figure 9a shows the equivalent circuit used to determine the optimized resistance and capacitance values from the EIS data.

In the equivalent circuit, *R_s_* is the solution resistance, *C_film_* is the capacitance of the passive film, *R_film_* is the electrical resistance of an ionic conduction path through the passive film, *C_dl_* is the double-layer capacitance and *R_ct_* is the resistance associated with the metal dissolved in the electrolyte [47]. Figure 9b,c show the experimental Nyquist plots (scatter plots) and calculated Nyquist plots using the equivalent circuit (line plots). The Nyquist plots in Figure 9 show two time constants, and the measurement plots and the calculation plots agree well. The semicircle in the low frequency is correlated with charge transfer resistance, and the high-frequency semicircle indicates the corrosion resistance by the passive film [48,49,50]. Since the real axis value at the low-frequency intercept is the sum of the polarization resistance and solution resistance, the corrosion resistance increases as the diameter of the semicircle increases [51,52]. The high-frequency semicircle of BM is bigger than that of FZ in both alloys. Therefore, the corrosion resistance of FZ was lower than that of BM due to the effect of passive film stability. Additionally, as a result of analyzing the polarization resistance (*R_p_*), which means the diameter of semicircles, using an equivalent circuit, in the case of BM, that of AA3003 was 17,748 Ω·cm^2^ and that of U3003 was 28,692 Ω·cm^2^. In the case of FZ, the polarization resistance of AA3003 was 2250 Ω·cm^2^ and that of U3003 was 11,993 Ω·cm^2^. Therefore, the corrosion resistance of U3003 was higher than that of AA3003 in FZ as well as BM.

The EIS fitting results are presented in Table 5.

The corrosion rate (*i_corr_*) and polarization resistance (*R_p_*) are inversely proportional, as shown in Equation (1) [53],
(1)$$i_{corr}=RT/zFR_{p}$$
where *R* represents the gas constant (8.314 J/mol·K), *T* represents temperature, *F* represents Faraday’s constant (96,500 C/mol), and *z* represents the number of equivalents exchanged. *R_p_* represents the polarization resistance produced by the sum of *R_film_* and *R_ct_* because two resistors connected in series [54]. As in the immersion test, the corrosion rate of FZ was faster than that of BM in both alloys. In the case of AA3003, the corrosion rate of FZ (3.80 µA/cm^2^) increased by 7.9 times compared to that of BM (0.48 µA/cm^2^), and in the case of U3003, the corrosion rate of the FZ (0.71 µA/cm^2^) increased by 2.4 times compared to that of BM (0.30 µA/cm^2^).

In both alloys, the *R_film_* of FZ was lower than that of BM, and the *C_film_* was higher. This means that the thickness of the passive film of FZ became thinner and unstable [55,56]. According to Equations (2) and (3), the thickness of the passive film (*d_film_*) is proportional to *R_film_* and inversely proportional to *C_film_* [48,57].
(2)$$d_{film}=R_{film}A/\rho$$
(3)$$d_{film}=\varepsilon \varepsilon_{0}A/C_{film}$$
where *A* represents the surface area, *ρ* represents the specific resistivity of the passive layer, *ε* represents the permittivity of passive film, and *ε*_0_ represents the permittivity of the vacuum (8.85 × 10^−12^ F/m). After laser welding, the IMCs were evenly distributed on the surface of the FZ, making the passive film thin and unstable [58,59,60]. As shown in Figure 8d, the IMCs appeared in the passive film, which acted as defects. Consequently, the weakening of the passive film induced an increase of corrosion rate of the FZ of both alloys.

The summary of the corrosion mechanism of laser welded Al–Mn alloys is schematically shown in Figure 10.

After laser welding, the IMCs were refined and dispersed in the FZ due to interdendritic segregation. This dispersion of IMCs increased the area of IMC to the matrix, and destabilized the passive film. As a result, the corrosion rate of the FZ increased in both alloys. However, in comparison to AA3003, U3003 exhibited a smaller IMC size and reduced quantity, along with a lower Mn/Fe ratio. Therefore, U3003 was more resistant to micro-galvanic corrosion between the IMC and the Al matrix. Consequently, even subsequent to the laser-welding process, U3003 demonstrated higher corrosion resistance than AA3003.

## 4. Conclusions

In this study, the corrosion behavior of a laser-welded Al–Mn–Zr alloy for a heat exchanger was investigated and compared with that of AA3003. According to the immersion test and the EIS test, the corrosion rate of AA3003 was faster than that of U3003 in both FZ and BM. During the laser welding, interdendritic segregation was occurring in the FZ, and the Al_6_(Fe,Mn) IMC was refined and spread more evenly. This change in IMC distribution destabilized the passive film and increased the micro-galvanic corrosion rate. However, the addition of Zr suppressed the precipitates of IMCs, and the remaining Al_3_Zr made the IMC spherical in the FZ. Therefore, the IMC area ratio to the Al matrix did not change significantly and the effect of micro-galvanic corrosion was negligible in U3003. Consequently, U3003 had higher corrosion resistance than AA3003 even after laser welding. Therefore, using U3003 as a material for fins and tubes can improve the corrosion resistance of laser-welded joints of the fins and tubes in heat exchangers.

## Figures and Tables

**Figure 1 materials-16-06009-f001:**
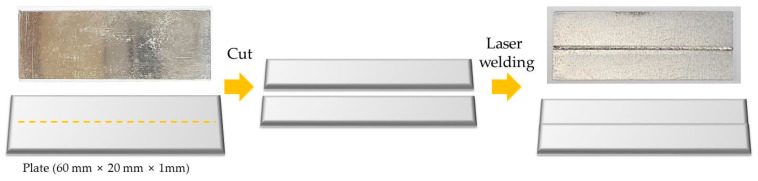
Procedure for laser-welded specimen preparation.

**Figure 2 materials-16-06009-f002:**
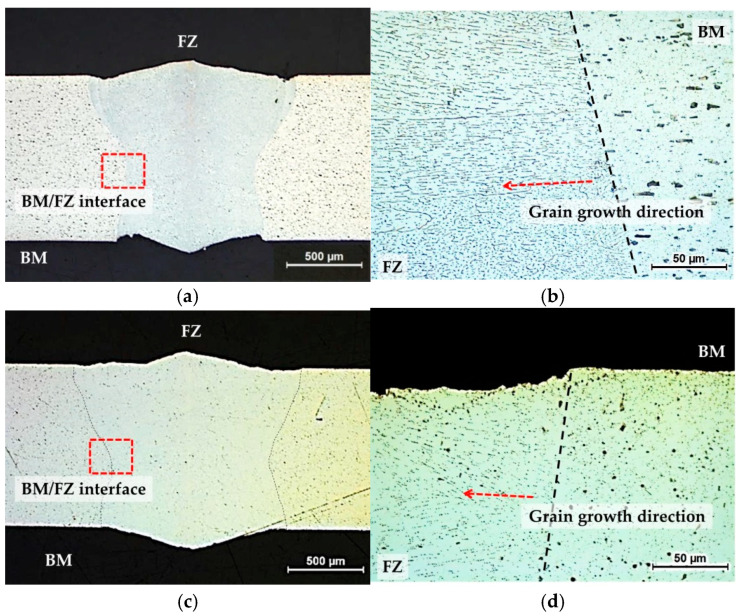
Cross-sectional images of laser-welded specimens. AA3003 (**a**) 50× and (**b**) 500× (BM/FZ Interface); U3003 (**c**) 50× and (**d**) 500× (BM/FZ interface).

**Figure 3 materials-16-06009-f003:**
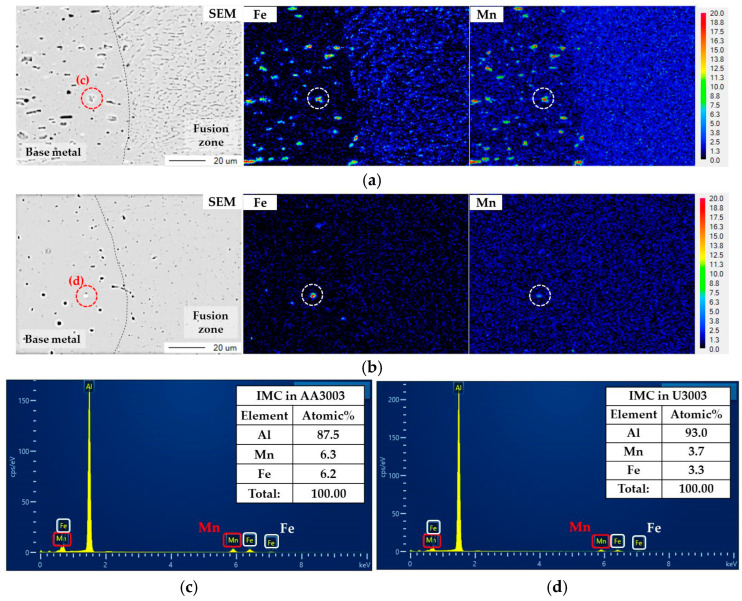
SEM images of BM/FZ interface and EPMA element mapping for Fe and Mn of (**a**) AA3003 and (**b**) U3003. The same position as the red circle in the SEM image is indicated by a white circle in the mapping image. EDS point analysis results of IMCs in red circles of (**c**) AA3003 and (**d**) U3003.

**Figure 4 materials-16-06009-f004:**
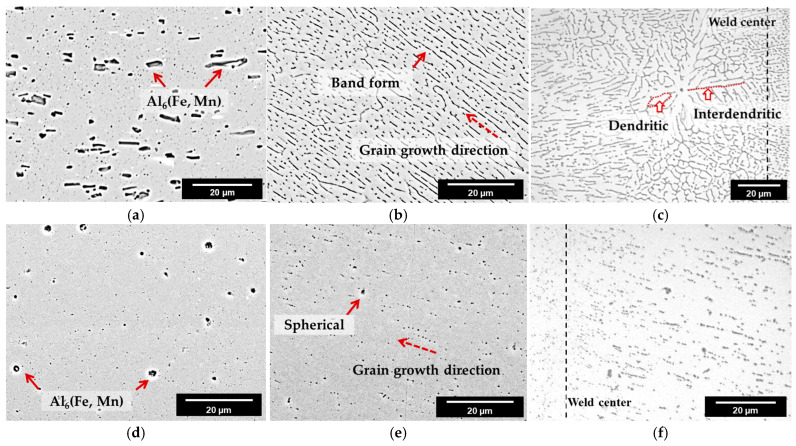
Microstructures of laser-welded specimens. AA3003 (**a**) BM, (**b**) FZ, near BM/FZ interface, and (**c**) FZ, weld center; U3003 (**d**) BM, (**e**) FZ, near BM/FZ interface, and (**f**) FZ, weld center.

**Figure 5 materials-16-06009-f005:**
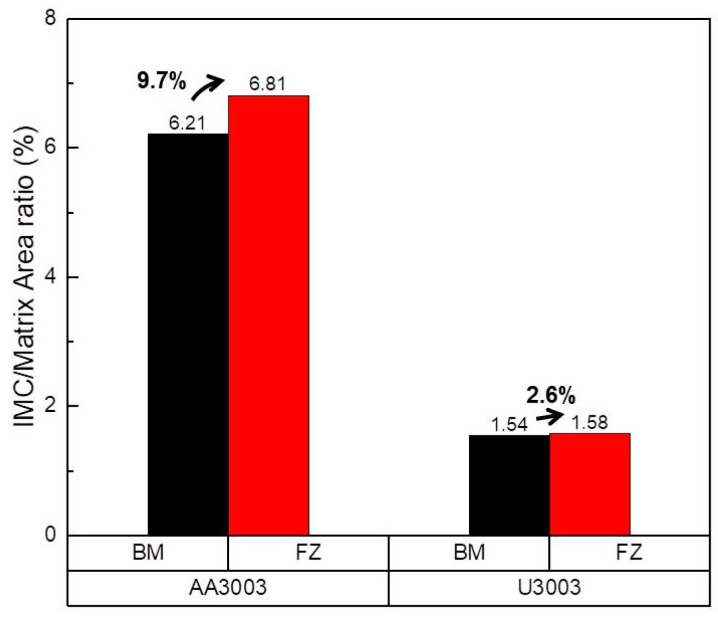
The area ratio of IMCs to Al matrix in BM and FZ calculated from SEM images in Figure 4.

**Figure 6 materials-16-06009-f006:**
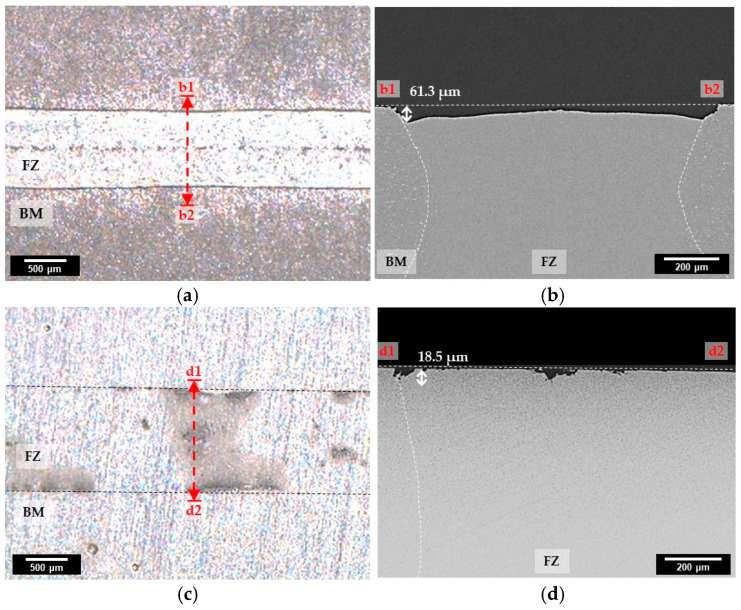
Surface and cross-sectional images of specimens after 12 weeks’ immersion test. (**a**) Surface image of AA3003 and (**b**) cross-sectional image at the b1–b2. (**c**) Surface image of U3003 and (**d**) cross-sectional image at the d1–d2.

**Figure 7 materials-16-06009-f007:**
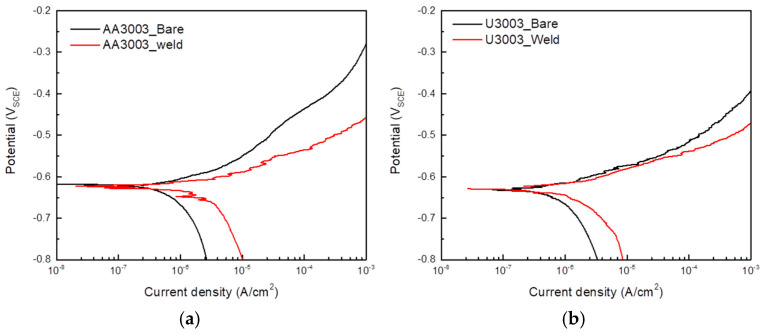
Potentiodynamic polarization curves of (**a**) AA3003 and (**b**) U3003.

**Figure 8 materials-16-06009-f008:**
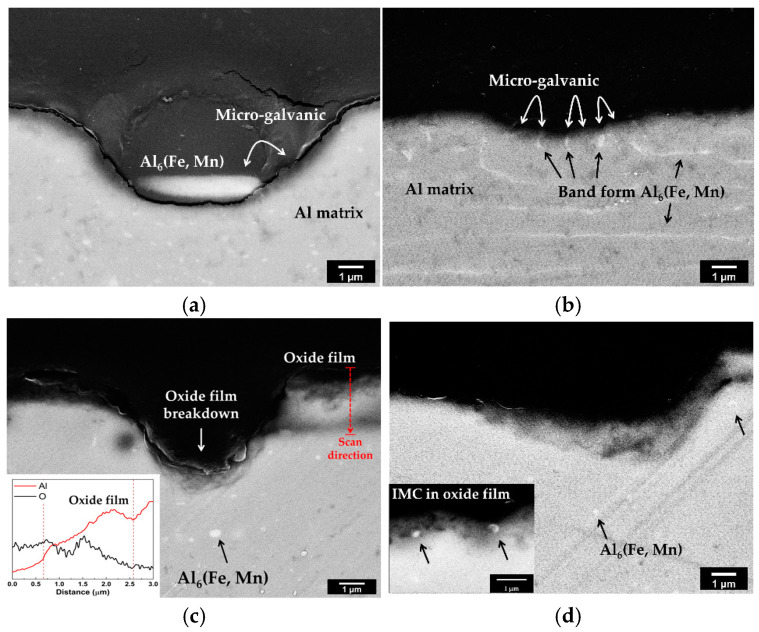
High-magnification images of cross sections of specimens after 12 weeks’ immersion testing. AA3003 (**a**) BM and (**b**) FZ; U3003 (**c**) BM and (**d**) FZ.

**Figure 9 materials-16-06009-f009:**
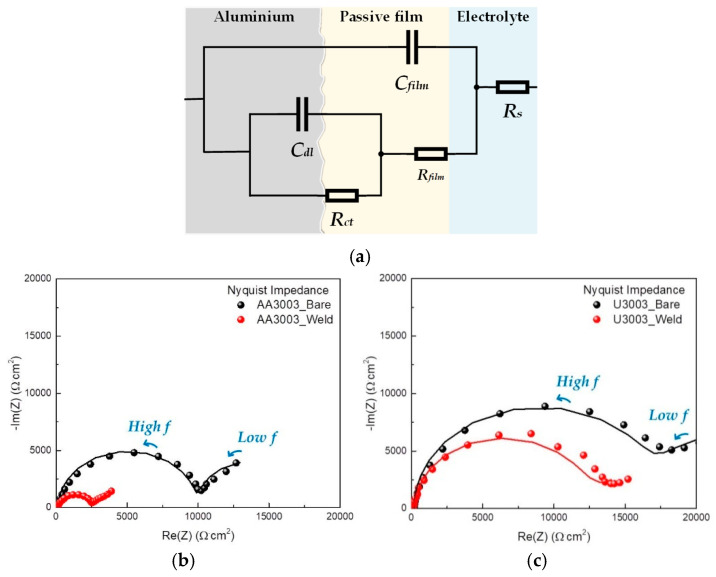
(**a**) Equivalent circuit to fit the EIS data. EIS Nyquist plots of (**b**) AA3003 and (**c**) U3003. Scatter plots are experimental values and line plots are calculated values.

**Figure 10 materials-16-06009-f010:**
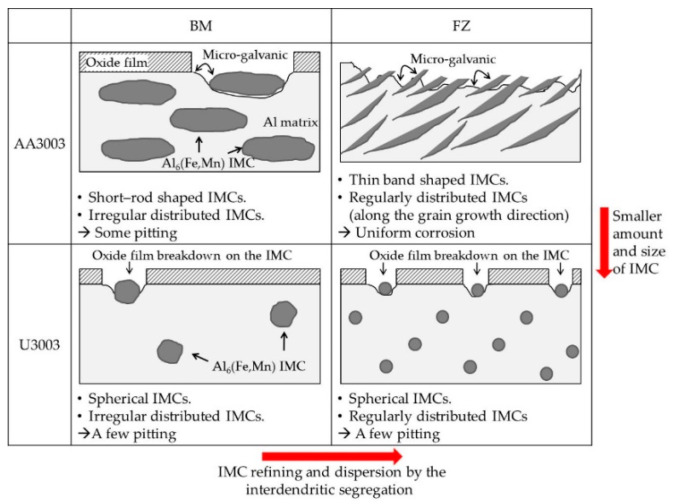
Schematic mechanism of corrosion behavior of laser-welded Al–Mn alloys.

**Table 1 materials-16-06009-t001:** Chemical compositions of Al alloys.

Alloys	Composition (wt.%)						
	Mn	Fe	Zr	Si	Cu	Mg	Al
AA3003	1.126	0.357	-	0.100	0.120	0.017	Bal.
U3003	0.485	0.086	0.164	0.032	-	0.002	Bal.

**Table 2 materials-16-06009-t002:** Laser welding parameters.

Power Mode	Speed (m/min)	Beam Power (kW)	Beam Size (μm)	Wavelength (μm)	Incident Angle (°)	Defocusing (mm)
Continuous	10	Main 3.0 Ring 0.5	Main 210 Ring 540	1070	0	0

**Table 3 materials-16-06009-t003:** Chemical composition of artificial acid rain.

pH	SO_4_^2−^ (mg/L)	NO^3−^ (mg/L)	Cl^−^ (mg/L)
4	2.212	1.545	400

**Table 4 materials-16-06009-t004:** Potentiodynamic test results for AA3003 and U3003.

Alloy	AA3003-Bare	AA3003-Weld	U3003-Bare	U3003-Weld
Corrosion potential (mV_SCE_)	623.7	621.7	631.2	631.2
Corrosion current density (μA/cm^2^)	0.78	3.15	0.46	0.69

**Table 5 materials-16-06009-t005:** EIS analysis results for AA3003 and U3003.

Alloy		*R_s_* (Ω·cm^2^)	*C_film_* (F/cm^2^)	*R_film_* (Ω·cm^2^)	*C_dl_* (F/cm^2^)	*R_ct_* (Ω·cm^2^)	*i_corr_* (µA/cm^2^)
AA3003	BM	144.12	7.39 × 10^−6^	9840	1.7 × 10^−3^	7908	0.48
	FZ	86.2	10.3 × 10^−6^	404	0.01 × 10^−3^	1846	3.80
U3003	BM	181.62	6.6 × 10^−6^	17,622	0.3 × 10^−3^	11,070	0.30
	FZ	117.42	11.2 × 10^−6^	993	0.6 × 10^−3^	11,000	0.71

## Data Availability

Data is contained within the article material.

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
