# Peer review of "Microstructure and Corrosion Behavior of Laser-Welded Al–Mn–Zr Alloy for Heat Exchanger"

_materials, 2023, doi:10.3390/ma16176009_

Round 1

Reviewer 1 Report

In this manuscript, the microstructure and corrosion behaviour of laser welded Al-Mn-Zr is assessed and compared to the equivalent laser welded Al-Mn alloy. According to the results, Zr addition improves the corrosion resistance of the welded material in artificial acid rain. This work is interesting and should be considered for publication in materials. Please find my comments below:

1) Authors need to explain the novelty of this work. Why they decided to study the addition of Zr in Al-Mn?

2) Authors analysed the microgalvanic behaviour between different phases. They need to explain the galvanic corrosion between areas with different microstructural features and chemical composition (FZ-BM).

3) Authors need to add more SEM images of higher magnification to support the discussion on the selective dissolution and micro-galvanic corrosion.

4) Potentiodynamic polarisation curves would help understand the differences in the corrosion performance of the two different configurations and expand the discussion.

5) It would be beneficial to add relevant results on the corrosion performance of bulk Al-Mn and Al-Mn-Zr alloys. Are there any differences compared to the welded material?

Author Response

Thank you for your kind letter. Please see the attachment.

Reviewer 2 Report

The information reported in the manuscript is interesting, however, some aspects must be revised and in lines 104 and 105 authors mentioned that: Before conducting EIS, all working electrodes were immersed in the test solution for 30 hours to attain equilibrium of the specimen surface. The corrosion of any metal immersed in an electrolyte is not an equilibrium process, then the surface will never achieve an equilibrium potential, that must be changed to reach a steady state potential.

In the experimental section authors must indicate the times the EIS measurements were conducted for each alloy. Also please explain the reason for the 12 weeks immersion period

The images in Figure 4 must be improved to show clearly the interdendritic distribution of the IMCs. 

In line 2555, please explain what is Qfilm.

Lines 273-275 must be rewritten, they make no sense.

Authors must revise carefully the English language because there are some lines that make no sense.

Author Response

(The authors gave the same response as above.)
